# Quantifying Spatial Heterogeneity of Tumor-Infiltrating Lymphocytes to Predict Survival of Individual Cancer Patients

**DOI:** 10.3390/jpm12071113

**Published:** 2022-07-07

**Authors:** Aleksandra Suwalska, Lukasz Zientek, Joanna Polanska, Michal Marczyk

**Affiliations:** 1Department of Data Science and Engineering, Silesian University of Technology, 44-100 Gliwice, Poland; aleksandra.suwalska@polsl.pl (A.S.); joanna.polanska@polsl.pl (J.P.); 2Faculty of Automatic Control, Electronics and Informatics, Silesian University of Technology, 44-100 Gliwice, Poland; zientek.luk@gmail.com; 3Yale Cancer Center, Yale University, New Haven, CT 06510, USA

**Keywords:** TILs, spatial measures, histopathological images, survival, cancer prognosis

## Abstract

Tumor-infiltrating lymphocytes (TILs), identified on HE-stained histopathological images in the cancer area, are indicators of the adaptive immune response against cancers and play a major role in personalized cancer immunotherapy. Recent works indicate that the spatial organization of TILs may be prognostic of disease-specific survival and recurrence. However, there are a limited number of methods that were proposed and tested in analyses of the spatial structure of TILs. In this work, we evaluated 14 different spatial measures, including the one developed for other omics data, on 10,532 TIL maps from 23 cancer types in terms of reproducibility, uniqueness, and impact on patient survival. For each spatial measure, 16 different scenarios for the definition of prognostic factor were tested. We found no difference in survival prediction when TIL maps were stored as binary images or continuous TIL probability scores. When spatial measures were discretized into a low and high category, a higher correlation with survival was observed. Three measures with the highest cancer prognosis capability were spatial autocorrelation, GLCM M1, and closeness centrality. Most of the tested measures could be further tuned to increase prediction performance.

## 1. Introduction

Nowadays, we observe the rapid worldwide growth of mortality due to cancer, mostly because of the aging population, and the burden of cancer will probably intensify much more. Early and accurate prediction is the key to recovery and many efforts have been taken to improve this process, but appropriate and efficient cancer treatment could also lead to a significant increase in cancer patient survival. While there are many types of treatments for different cancers, including surgery with chemotherapy and/or radiation therapy, targeted therapy, or hormone therapy, immunotherapy that stimulates or suppresses the immune system to help the body fight cancer has recently received much more attention. Unfortunately, the immune responses following the same treatments may be specific to each individual patient [1], which increases the importance of precision medicine in this field. Increasing evidence indicates that interactions between tumor cells, tumor stroma, and the tumor immune microenvironment could also evolve during the disease and could impact the response to therapies [2]. Especially important are lymphocyte cells that have a huge role in the immune-system response to many inflammatory diseases, including cancer.

Whole-slide images (WSIs) of cancer tissues contain a significant portion of diagnostic and prognostic data, which can be efficiently extracted by computational methods in a quantitative way to support fast and accurate clinical decisions. Important tissue structure and different immune-system cells are usually quantified by a pathologist’s visual inspection of hematoxylin and eosin (HE)-stained slides. Manual investigation of large HE images is slow and could be imprecise without any software assistance [3]. Computational profiling could eliminate known problems of interrater reliability. Tumor-infiltrating lymphocytes (TILs), identified in the close vicinity of the tumor and between the tumor cells on HE images, are indicators of the adaptive immune response against tumors and play a key role in cancer immunotherapy [4]. Increased TIL concentration can predict response to chemotherapy and is associated with a survival benefit in breast cancer [5], non-small-cell lung cancer [6], melanoma [7], or esophageal adenocarcinoma [8]. In breast cancer treatment, assessment of TILs should be included in clinical guidelines very soon [5,9].

Recently, it has been noticed that there exists an association between the spatial context of TILs and the impact on cancer patient survival [10,11]. Positions of lymphocytes detected on HE images can be used to construct maps of TILs. Then, the maps could be evaluated to measure the spatial heterogeneity of TILs. Unfortunately, in the literature, there are a limited number of methods to analyze the spatial structure of TILs that were already tested. In this work, we collected and evaluated 14 different measures of spatial heterogeneity (termed spatial measures), including the one previously applied to TIL-map analysis, and also methods developed for the analysis of imaging mass spectrometry or spatial transcriptomics data. We tested these measures on TIL maps from 23 cancer types created elsewhere [12] with the ultimate goal to find the most robust and efficient indices that could be used for cancer prognosis.

## 2. Materials and Methods

### 2.1. Data

The analysis was carried out on the spatial maps of lymphocytic infiltrates generated from the images of hematoxylin and eosin-stained tissue slides from The Cancer Genome Atlas (TCGA) in a previous study [12]. The TIL maps were obtained by the use of convolutional neural networks on images divided into small patches (50 × 50) and trained for tumor-infiltrating lymphocyte detection [12]. In the original paper [12], the information about TIL was stored using two TIL map scales: (i) binary (if there are TILs on the patch or not); (ii) probability (values from 0.5 to 1; closer to 1 means a higher chance of TILs; values smaller than 0.5 means that there are no TILs on the patch). These TIL maps were made publicly available by the authors. In total, 7776 TIL maps generated for 7426 patients representing 23 different cancer types were analyzed in this study (Table 1).

### 2.2. Data Preprocessing

The preprocessing of raw TIL maps was performed in 3 steps: (i) Detection and removal of dark lines (artifacts); (ii) separation of non-touching dark TILs areas into regions of interest (ROIs); (iii) removal of background artifacts. In addition, images with fewer than 10,000 pixels (e.g., 100 × 100) were removed prior to analysis since normally WSIs, which are analyzed in this study, are much bigger.

In the first step of data preprocessing (artifact lines removal), the total number of pixels that were marked as TILs on a TIL map (pixel value > 0) was calculated separately for each row and column of the image. Then, using a second-derivative method, the number of TIL-marked pixels was compared between adjacent rows and columns. The line representing artifact was detected and removed if the percentage of TIL-marked pixels in particular row/column was higher than 20% of image width/height in comparison to adjacent rows/columns (Appendix A).

To define the regions of interest (ROIs; Figure 1 and Appendix A), TIL map was first binarized into regions with and without TILs and only regions with TILs were selected. Unimportant regions with an area smaller than 10% of the total area of the tissue were marked as background artifacts and removed. Before saving, all ROIs were cropped (removing white background around tissue) to maximize the TIL region area on the image (Appendix A).

### 2.3. Methods for Spatial Structure Analysis (Spatial Measures)

Fourteen different measures of spatial heterogeneity of TILs calculated on medical HE images were tested. Some methods were already used in the analysis of TIL maps, but others were developed for different types of molecular data.

#### 2.3.1. Methods Developed for Analysis of Spatial Molecular Data

The first group of methods is implemented in Squidpy software, and in general allows for the identification of spatial patterns in tissue [13]. Squidpy is a Python-based framework developed for spatially resolved omics data analysis. The following steps needed to be carried out to allow an analysis of TIL maps. First, each TIL map was read into the AnnData object that stores annotated data matrices in a form of coordinates of the tissue pixels and two feature columns: probability and binary decision of being a TIL. The column indicating the binary assignment was also used as a cluster indicator dividing the data into TILs and other tissue. In Squidpy, spatial information is encoded using spatial graphs, and description and quantification of spatial patterns are performed using several metrics including Ripley’s F, G and L, spatial autocorrelation, and centrality scores (degree centrality, closeness centrality, average clustering). Ripley’s statistics were taken for the search radius equal to 50. The rest of the statistics were calculated for each cluster, but we maintained the values only for the TILs cluster.

#### 2.3.2. Clustering-Based Methods

In [10], several spatial measures based on affinity propagation (AP) clustering were introduced. Affinity propagation identifies a subset of representative examples called exemplars to detect patterns in data [14]. It does not require the number of clusters to be defined before beginning the calculations, which can be a huge benefit and a huge downfall at the same time; for large chunks of data, where data-point accumulation is dense, the algorithm requires a lot of time to finish its calculations. Clustering results were then utilized to estimate spatial heterogeneity using the following measures: Ball–Hall index, Banfeld–Raftery index, C-index, and determinant ratio from clusterCrit R package. Ball–Hall index is a method for data analysis and pattern classification. It is defined as the mean of the dispersion across all the clusters which is equivalent to the squared distances of the points of the cluster considering its center. Banfield–Raftery index is defined as the weighted sum of logarithms of the traces of the variance–covariance matrix across all the clusters. A detailed description of all other metrics could be found in clusterCrit R package vignette.

#### 2.3.3. Other Methods

Two other methods that originate from the analysis of imaging mass spectrometry data were tested: a grey level-occurrence matrix-based method and spatial chaos.

A grey level-occurrence matrix (GLCM) is a second-order statistical texture-analysis method for texture-feature extraction [15]. The method was implemented in EXIMS software [16]. They assumed that structured images should have a pattern that contains a large number of pairs of pixels with co-occurring low intensity values and co-occurring high intensity values (the contrast between those two sets of intensity values shows a clear structure) [16]. The method starts by reducing the number of intensity levels to 8 and then the standard procedure of GLCM is carried out. After that, the weights can be introduced. The weights are different for each element of the GLCM matrix and are used to obtain two measures, M1 and M2, which stand for areas (regions) with low and high intensities. 

Spatial chaos (SC) is a method for measuring the spatial structure of ion-intensity maps [17]. It was introduced as solution to problems that occur while analyzing the MALDI-IMS data sets, i.e., detecting unknown molecules and testing for the presence of known molecules. The goal of the method is to rank the level of structure on the image. The high SC is defined as a lack of spatial pattern in the pixel intensities [17]. For each spectral feature the ion-intensity map is created and a two-step edge-detection filter for noisy images is applied to detect signal-intensity edges. Next, a one-nearest-neighbor graph on edge pixels is built. The measure of chaos is calculated on mean length of the graph edges. For images consisting of spatially connected structures (images displaying spatially structured intensity patterns) the value of SC is low; for images with spatially chaotic pixel intensities the value is high.

### 2.4. Survival Analysis

To score the quality of the chosen spatial measures in terms of accurate estimation of overall survival (OS) or progression-free interval (PFI), Cox regression was calculated both for continuous and discretized (low/high) data. To dichotomize continuous predictors in survival analysis, a minimum *p*-value approach was used [18]. The method was developed to select a cut-point with a maximum *χ*^2^ statistic when the outcomes are binary, but was extended to also analyze the survival outcomes [19]. Briefly, for each spatial measure, a set of 30 threshold values was estimated between 10 and 90 percentiles of spatial measures scaled to 0–1 values. Feature values lower than the threshold were changed to 0 and values higher than the threshold were changed to 1. For each threshold, the Cox regression model was created, and the resulting p-value was stored. We set the penalized parameter of the CoxPHFitter class to 0.1, which attaches a penalty term to the regression in order to improve the stability of the estimates. Since there were 16 different scenarios of survival analysis for 14 spatial measures in 23 cancer types, we were not able to find the one optimal value of the penalized parameter. The selected value was a good compromise for all cases. The final division was obtained with the threshold for which the lowest p-value was received. In each Cox regression model, patient age was added as a covariate. Two types of Cox models were then examined: (i) original, including spatial measure and age covariate; (ii) adjusted, extended with percentage of TILs as an additional covariate. All 16 possible scenarios of survival analysis are listed in Appendix A. The Kaplan–Meier graphs were created on discretized data for OS as well as for PFI to present the quality of the predictors and their influence on the survival. Infinity or NA values of the spatial measures were dropped prior to analysis. 

### 2.5. Statistical Analysis

A robust version of the coefficient of variation (CV) was used in this study. Strictly, CV was calculated by dividing median absolute deviation by sample median. Finally, the absolute value of obtained CV was used. The correlation between variables was estimated using both Pearson and Spearman sample correlation estimators. All two-group comparisons were performed using a non-parametric Mann–Whitney U test. For all tests, the statistical significance level was set to 0.05. Two-sided tests were performed in all cases.

## 3. Results

### 3.1. Definition of Region-of-Interest (ROI)

The TIL maps downloaded from the publicly available repository contained a significant amount of very small regions that were probably an artifact of background noise left after thresholding (Figure 1 and Appendix A). In addition, additional line artifacts were present. We introduced a few data preprocessing steps to remove these small regions and other artifacts and define ROIs on which spatial measures could be calculated. In the analyzed images we found two different situations: (i) different tissue fragments were present on the same slide (Figure 1A); (ii) two adjacent slices of the same tissue fragment were put on the same slide (Figure 1B). After data preprocessing, from a single TIL map, one to three regions were created, increasing the total number of analyzed images by 35% to 10,532 (Table 1). However, the total size of the images was reduced by cropping the background area around the tissue.

### 3.2. Different Representation of TIL Maps

In the original paper [12], TIL maps were stored in two scales: (i) binary, including only information of whether TILs are present or not; (ii) probabilities from deep-learning models, with values ranging from 0.5 to 1 (patches with values lower than 0.5 were not counted as TILs). For seven metrics, excluding spatial measures calculated with Squidpy software, we obtained separate values for these two formats. In some cases, it was not possible to calculate the exact value of some spatial measures since NA or infinity values were estimated (Appendix A). This problem was mostly observed for TIL maps with probabilities, where for two cancer types (meso and paad) no values could be estimated for three measures: GLCM M1, GLCM M2 and determinant ratio.

We first compared the values of spatial measure between two TIL map scales, binary and probabilities, by calculating the Spearman correlation coefficient (Figure 2). The highest correlation (close to 1 for each cancer type) was found for the Banfeld–Raftery index; however, in some individual cases, the values calculated on binary TIL maps were much lower than the one calculated on TIL maps with probabilities. GLCM M2 and spatial chaos showed the lowest correlation. For the spatial chaos measure, there is a group of TIL maps for which the same value was calculated on both TIL map scales (Figure 2B; diagonal line), but in some cases we observed much lower spatial chaos values calculated on TIL maps with probabilities.

### 3.3. Reproducibility of Spatial Measures across ROIs and Variation across Patients 

Since there was more than one ROI for some groups of TIL maps, we were able to establish the reproducibility of each spatial measure across ROIs by calculating the robust coefficient of variation (CV) separately for binary TIL maps and the one with probabilities (Figure 3A,B). The highest average CV was observed for the determinant ratio for both TIL map scales, where in some cases, huge differences between ROIs of the same patient were observed (CV > 10). The lowest CV was observed for Ripley G; however, in most situations, CV was equal to 0, meaning the same value for all ROIs. Other methods showed similar levels of reproducibility. In the subsequent analysis, the spatial measures for the ROIs that belonged to the same patient were averaged. We did not weight the measures by the ROI size during averaging, since we assumed that they should be independent of the ROI size. If the spatial colocalization is the same on the bigger and smaller tissue fragment, a good spatial measure should give the same value.

To test if spatial measures are different enough between patients, to represent different survival rates after treatment, we calculated the CV across patients (Figure 3C,D). Ripley’s F and G indices gave the same values for each patient within each cancer type. In addition, the average clustering measure was similar between patients. The highest variability within each cancer type was observed for the GLCM M2, Banfield–Raftery, and determinant ratios. For uvm cancer and TIL maps with probabilities, the CV of spatial chaos measure could not be estimated, because the median value across patients was equal to 0.

Finally, we calculated the correlation between spatial measures for all cancers and TIL map scales (Appendix A). We found that there is a strong positive linear correlation between two cluster centrality scores—degree centrality and closeness centrality—and also between two GLCM-based measures (Appendix A, upper triangle). Other associations were rather nonlinear (Appendix A, lower triangle). We also observed a strong positive correlation between Ball–Hall, Banfeld–Raftery, and C-Index, and strong negative correlation between Banfeld–Raftery and the determinant ratio.

### 3.4. Survival Prediction by Different Measures in Cancer Patients

Due to low reproducibility or low variation across patients within each cancer type, the following measures were not used in survival analysis: Ripley F, Ripley G, average clustering, and determinant ratio. Furthermore, the Banfeld–Raftery index was used only for TIL maps with probabilities. On the remaining 10 measures, we ran the survival analysis using original and TILs percent adjusted Cox model for two survival endpoints (OS and PFI), using continuous spatial measure scale and discretized into low/high groups (Appendix A).

Overall results of survival analysis showed that different spatial measures work well in different cancer types; however, there is no strong single candidate for the best measure for prediction of cancer patient survival (Figure 4). Across cancers, the highest correlation of spatial measures with survival was observed for tgct and thym for OS and cesc and uvm for PFI (Figure 4A,B). We found no significant difference in survival prediction performance when TIL maps were stored as binary images or continuous TIL probability scores (Appendix A). What was expected was that when the percent of TILs was added as a covariate in COX model, we obtained worse performance in survival models (Appendix A). There was a better performance when OS was used as a survival endpoint than PFI (Appendix A). Finally, when measures were discretized into a low and high category, a higher performance of survival models was observed (Appendix A).

Since no difference between TIL map scale was observed, difference between cancer types were analyzed only for binary TIL maps (Figure 4C,D, Appendix A). To find the best spatial measures, for each cancer, endpoint, and Cox model type, we ranked the measures based on Harrell’s concordance index. The three measures with the highest cancer prognosis capability resulting from averaged ranking were spatial autocorrelation, GLCM M1, and closeness centrality. Furthermore, all three measures had similar positions in ranking for different endpoints and Cox model types. 

Finally, skcm cancer type was chosen for further analysis (Figure 5), but all results of survival analysis were stored in Appendix A. For this cancer type, lower values of spatial chaos, Ripley L, spatial autocorrelation, and Banfeld–Raftery prognose longer survival. On the other hand, higher values of percent of TILs, degree centrality, and closeness centrality were associated with longer survival. Similar findings were obtained for survival analysis on continuous and discretized spatial measures, and in most cases were significant even after adjustment by percentage of TILs (Figure 5A,B).

## 4. Discussion

The obtained results show that patients with higher densities of TIL maps are among those with the longer survival, which further underlines that the correlation between the level of lymphocytes infiltration and the OS or PFI is true to a certain degree. The results also reflect that the TIL percentage as a predictor can be used with a limited success, the reason being that the TIL percentage might not be a sufficient factor as other information such as TIL localization, and the structures it forms could also affect the cancer survival; therefore, the percentage itself cannot be used as a robust and accurate predictor. What is noteworthy is that the higher infiltration does not necessarily mean that any type of structure was formed (e.g., lymphocytes that envelope the cancer cells). This further underlines the fact that that the simple density does not account for other factors that might be crucial to the correct prognosis. While for types such as brca, luad, and lusc, the percentage densities were comparable, in another tumor types such as paad the TIL densities were much lower; the dataset consisted mostly of tissues for which the percentage was below 5%.

Recent works have shown that spatial organization of TILS could be used as a prognostic factor of disease-specific survival and recurrence [20]. In early-stage non-small-cell lung cancer, they propose scores that capture density and spatial colocalization of TILs and tumor cells that can predict likelihood of recurrence [21]. Another pan-cancer study introduces the maps of TILs that are created with a convolutional neural network, used for calculating associations of TIL local spatial structures with cancer type and survival [10]. In another work they examined the relationship between the global abundance and spatial features of TIL infiltrates with clinical outcomes and showed that large aggregates of peritumoral and intratumoral TILs were associated with the longer survival, whereas the absence of intratumoral TILs was associated with increased risk of recurrence [22]. Another measure was developed to characterize the spatial architecture patterns of TILs together with surrounding cells for HPV-associated oropharyngeal squamous cell carcinoma and was associated with DFS in low-risk patients [23]. ArcTIL was created for quantitative characterization of the architecture of TILs and their interplay with cancer cells in three different gynecological cancers [24]. Most of these measures were developed exclusively for a given cancer; here, we review measures in terms of universal prediction models for multiple cancers. In addition, the methods described above are much more complicated and might include additional covariates or clinical features to be able to be calculated, while we assumed that TIL maps should be enough. 

In this work, we explored the clinical significance of 14 spatial measures of TIL maps in multiple cancer types to help guide future treatment. Not all indices were significant for all cancer types and the performance of survival models changed with each type of tumor. Among the best prognostic factors, we found two measures from the Squidpy package that were originally developed for analysis of spatial transcriptomics data, and one measure originating from analysis of imaging mass spectrometry data (GLCM M1). Thus, we proved that probably any spatial measure could be adapted for TIL-map analysis. Interesting results were achieved for both measures based on GLCM matrices. In these methods, there are multiple parameters (e.g., weight of pixels used during summarizing) that could be easily tested and tuned to increase their efficiency in TIL-map analysis.

The study has some limitations. First, in the analysis we used TIL maps that were provided by others [12]. In the original study, they showed high concordance of TIL maps with expert annotations; however, they might not be perfect in some cases. The spatial measures calculated on different TIL maps might bring slightly different survival results; however, we assume that the average overall ranking should be similar. Another limitation is that for some measures it is possible to obtain better results by tuning their parameters; here, we used default values. The next limitation is that survival results depend on the method used for dichotomization of each measure into low/high groups, and the minimum p-value algorithm used here might not be optimal in all cases. Lastly, in some WSIs, consecutive cuts of the same tissue were present, while in the others, different tissue fragments were stored. We expect that a higher reproducibility should be found when consecutive cuts are analyzed as opposed to different fragments, but due to lack of information on which ROIs are consecutive cuts, we could not check it.

We performed a comprehensive evaluation of 14 spatial measures, including different representation of TIL maps, two Cox model types, two survival endpoints, and two scales of spatial measures in 23 cancer types. We expect that the use of methodologies developed in this study will guide future researchers and ultimately will lead to expanding the knowledge on differences in TIL composition between different phenotypes. Sufficiently precise TIL-based signatures of high metastasis risk or poor prognosis for cancer patients will support the personalized treatment planning, minimizing the social and economic cost of cancer, and maximizing the patients’ comfort of life in the future.

## Figures and Tables

**Figure 1 jpm-12-01113-f001:**
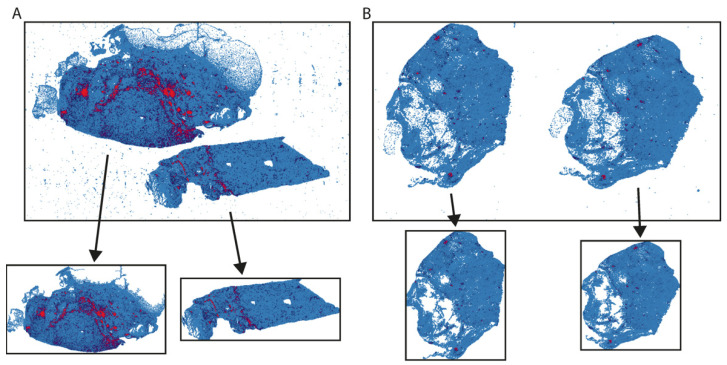
Example of TIL maps representing images consisting of two different tissue fragments present on the same slide (**A**) or two adjacent slices of the same tissue fragment (**B**). Top rows show original TIL maps, while bottom rows are regions of interest (ROIs) that were provided after data preprocessing.

**Figure 2 jpm-12-01113-f002:**
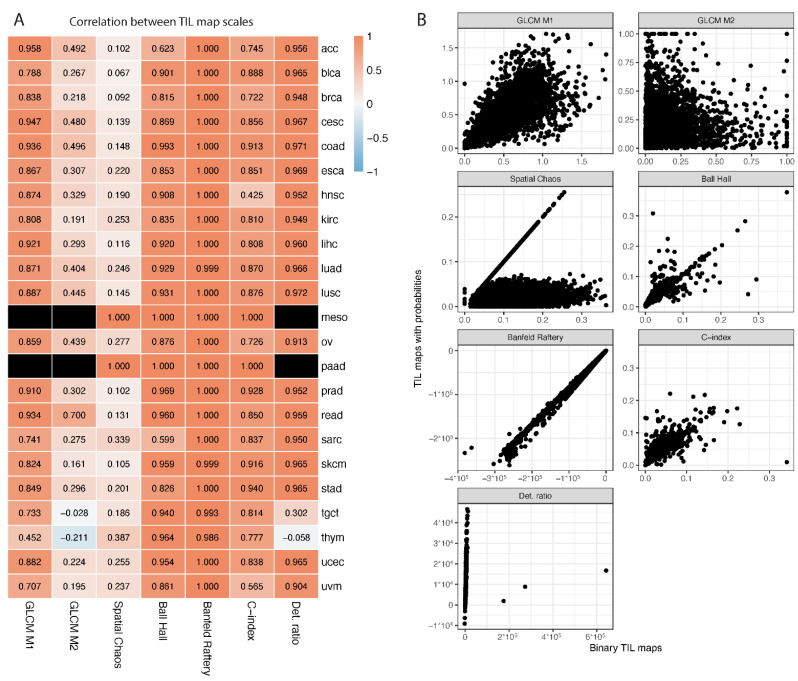
Comparison of spatial measures between two formats of TIL maps. (**A**) Spearman correlation of each spatial-measure value within each cancer type between binary and probability TIL maps. Black rectangles represent cases where NA values were present for all images within cancer type for at least one TIL map scale. (**B**) Scatter plots visualizing results for individual TIL maps within each spatial measure.

**Figure 3 jpm-12-01113-f003:**
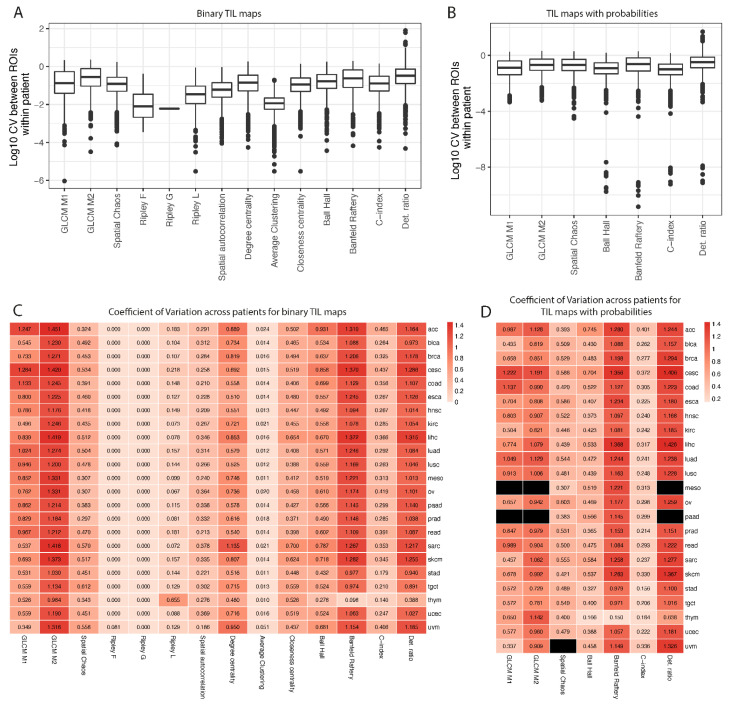
Variation of spatial measures across ROIs and patients estimated by robust coefficient of variation (CV). Boxplots shows reproducibility of measures calculated on different ROIs from the same patient for binary TIL maps (**A**) and probabilities (**B**). Heatmaps represent CV across patients for binary TIL maps (**C**) and probabilities (**D**). Black rectangles show situations where spatial measures could not be calculated or median across patients within the same cancer type was equal to 0.

**Figure 4 jpm-12-01113-f004:**
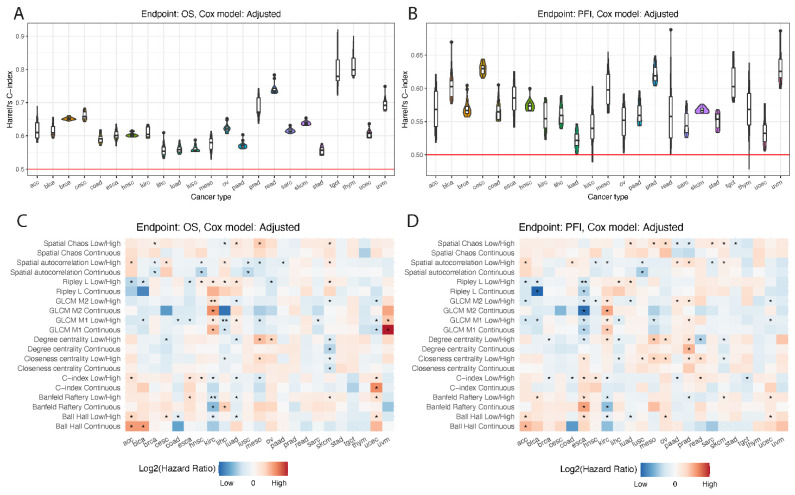
Result of survival analysis. Top: Distribution of Harrell’s concordance index from survival analysis across different cancer types for overall survival (**A**) and progression-free interval (**B**). Bottom: Heatmaps presenting hazard ratios per cancer type and spatial-measure scale (continuous or discretized) for overall survival (**C**) and progression-free interval (**D**). Single star indicates *p*-value lower than 0.05, and two stars lower than 0.01.

**Figure 5 jpm-12-01113-f005:**
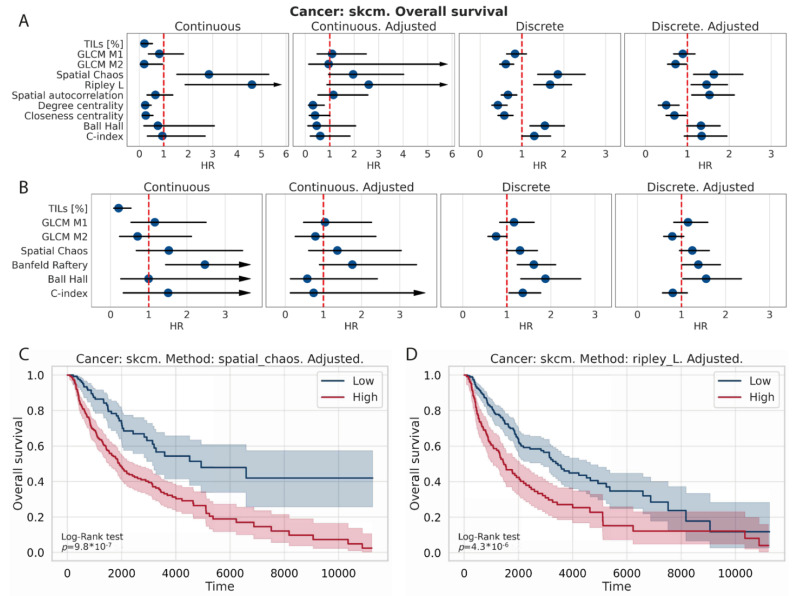
Detailed survival analysis results for skcm cancer type. Forest plots compare hazard ratios with confidence intervals between spatial measures for each Cox model type and measure scale separately for binary TIL maps (**A**) and probabilities (**B**). Kaplan–Meier curves of selected discretized spatial measures for adjusted model: spatial chaos (**C**) and Ripley’s L (**D**).

**Table 1 jpm-12-01113-t001:** Data description. TIL maps represent original images with TIL information, while regions of interest (ROIs) are tissue regions separated during the data-preprocessing step. One or more TIL maps were available per patient.

Cancer	Acronym	Patients	TIL Maps	ROIs
Adrenocortical carcinoma	acc	92	323	391
Bladder urothelial carcinoma	blca	179	179	223
Breast invasive carcinoma	brca	1067	1068	1312
Cervical squamous cell carcinoma and endocervical adenocarcinoma	cesc	268	268	526
Colon adenocarcinoma	coad	452	453	630
Esophageal carcinoma	esca	156	156	223
Head and neck squamous cell carcinoma	hnsc	450	450	698
Kidney renal clear cell carcinoma	kirc	513	514	626
Liver hepatocellular carcinoma	lihc	365	365	490
Lung adenocarcinoma	luad	479	480	662
Lung squamous cell carcinoma	lusc	484	484	655
Mesothelioma	meso	87	175	347
Ovarian serous cystadenocarcinoma	ov	106	106	180
Pancreatic adenocarcinoma	paad	183	189	253
Prostate adenocarcinoma	prad	403	403	548
Rectum adenocarcinoma	read	165	165	251
Sarcoma	sarc	255	255	316
Skin cutaneous melanoma	skcm	434	448	611
Stomach adenocarcinoma	stad	434	434	454
Testicular germ cell tumors	tgct	149	154	190
Thymoma	thym	121	121	152
Uterine corpus endometrial carcinoma	ucec	504	506	699
Uveal melanoma	uvm	80	80	95

## Data Availability

TIL maps were downloaded from the following website: https://stonybrookmedicine.box.com/v/til-results-new-model (accessed on 7 January 2022). Clinical data of TCGA patients used in survival analysis were downloaded from https://gdc.cancer.gov/about-data/publications/PanCan-Clinical-2018 (accessed on 7 February 2022). Computational pipeline developed for TIL maps processing, calculation of spatial measures and survival analysis is freely available for download from https://github.com/Aleksandra795/TIL_maps (accessed on 7 February 2022).

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
