# Peer review of "Quantifying Spatial Heterogeneity of Tumor-Infiltrating Lymphocytes to Predict Survival of Individual Cancer Patients"

_jpm, 2022, doi:10.3390/jpm12071113_

Round 1
Reviewer 1 Report
In this paper, authors generated a set of tumor-infiltrating lymphocyte (TIL) maps, and, from them, multiple features relating to spatial organization were extracted. Authors studied the correlation between such features in an inter and intra-patient fashion. Also, they generated Cox models for each feature and studied their association with overall and progression-free survival.
In general, the paper is well written although there are some sections that would benefit from rephrasing (see details below). In addition, the paper is technically sound, methodology appears reproducible, and results look promising. However, there are some issues that need to be addressed to make it suitable for publication:
· An important problem of the paper is that authors appear to suggest that p-value (or significance) is a performance metric. Here, it is important to mention that p-values are likely not comparable from different models, even with the same dataset. Unless authors are able to argue that p-value is an adequate metric here and include supporting references, other performance metrics must be used (e.g., Harrell’s concordance index). Here are some examples:
o “The final division was obtained with the threshold for which the lowest p-value was received.”
o “There was a slightly higher significance when PFI was used as survival endpoint than OS”.
o “When percent of TILs was added as a covariate in COX model, we obtained smaller significance of spatial measures”.
o “Not all indices were significant for all cancer types, but the significance was changing with each type of tumour.”
o Among others
· The analysis was carried out using images from TCGA. Were these frozen sections or diagnostic samples? If both were used, did authors find differences in spatial organization between them?
· Authors used a convolutional neural network to generate TIL maps. How was the performance of such a method in the employed dataset?
· As mentioned before, some sections would benefit from rephrasing. For example, the paragraphs 2 and 3 of Section 2.2 are not completely clear. For example, the paper says “Binarizing the TIL map into regions with and without TILs allowed to separate them into regions-of-interest”; this sounds confusing, does it mean that there are regions of interest for TILs and other for non-TILs? Authors are encouraged to include a figure (or adjust Figure 1) that clearly illustrates, step by step, the data processing. Similarly, paragraph 2 of section 3.2 is hard to follow and should provide more details. Also, in that paragraph, please indicate which correlation authors are referring to (Spearman’s or Pearson’s).
· The way authors discretized data seems a little unorthodox. Please include some references that back this approach.
· Statistical significance level was set to 0.05. Were p-values two-sided?
· According to Figure 1, some images contained consecutive cuts while others had different tissue fragments, and the experiment described in section 3.3 studied the reproducibility of spatial measures across ROIs. It sounds expected that there is correlation between such reproducibility and the type of ROI. A higher reproducibility should be found when consecutive cuts are analyzed as opposed to different fragments. Authors need to further discuss this point.
· The paper says that “three measures with the highest cancer prognosis capability were GLCM M1, Ripley’s L, and spatial autocorrelation, having similar position in ranking for different endpoints and Cox model types. Across cancers, the highest correlation of spatial measures with survival was observed for hnsc and skcm”. How did authors measure the prognostic capabilities of the models? If that is based in p-value, this may not be correct (See previous comment). Please clarify.
· Although the volcano plots look nice, they do not provide much information since the cancer type cannot be determined. The plots presented in Supplementary Figure 6 look more interesting in this sense, so they should be included in the main manuscript.
· Discussion should include some recent works in spatial analysis of TILs in different cancer types such as oropharyngeal and ovarian (See for example 10.1093/jnci/djab215 and 10.1136/jitc-2021-003833). Also, authors need to comment on the differences between their approach and the others existent in the literature. Finally, Discussion must include the limitations of the work.
Author Response
General comments: In this paper, authors generated a set of tumor-infiltrating lymphocyte (TIL) maps, and, from them, multiple features relating to the spatial organization were extracted. The authors studied the correlation between such features in an inter and intra-patient fashion. Also, they generated Cox models for each feature and studied their association with overall and progression-free survival. In general, the paper is well written although there are some sections that would benefit from rephrasing (see details below). In addition, the paper is technically sound, methodology appears reproducible, and results look promising. However, there are some issues that need to be addressed to make it suitable for publication.
Response: We are very grateful for all your comments and suggestions, and very happy that you found our manuscript interesting. Below, we responded to all your comments and upgraded the manuscript.
Remark 1: An important problem of the paper is that authors appear to suggest that p-value (or significance) is a performance metric. Here, it is important to mention that p-values are likely not comparable from different models, even with the same dataset. Unless authors are able to argue that p-value is an adequate metric here and include supporting references, other performance metrics must be used (e.g., Harrell’s concordance index). Here are some examples:
o “The final division was obtained with the threshold for which the lowest p-value was received.”
o “There was a slightly higher significance when PFI was used as survival endpoint than OS”.
o “When percent of TILs was added as a covariate in COX model, we obtained smaller significance of spatial measures”.
o “Not all indices were significant for all cancer types, but the significance was changing with each type of tumour.”
Response: That’s a great comment! We agree with the reviewer, that p-value is not a performance metric. We used it because we wanted only to point out the difference in statistical significance in different scenarios of survival analysis. But this might not be the best way for methods comparison, so as the reviewer suggested, we have calculated Harrell’s concordance index and added the following sentences into the text describing new results (section 3.4, second paragraph):
„We found no significant difference in survival prediction performance when TIL maps were stored as binary images or continuous TIL probability scores (Supplementary Figure 5A). What was expected, when the percent of TILs was added as a covariate in COX model, we obtained worse performance in survival models (Supplementary Figure 5B). There was a better performance when OS was used as a survival endpoint than PFI (Supplementary Figure 5C). At last, when measures were discretized into a low and high category, a higher performance of survival models was observed (Supplementary Figure 5D).”
The dichotomization of spatial measures into low/high groups for survival analysis was not changed in the revised version, however, we added the justification of our choice (see the response to Remark 5).
Remarks 2 & 3: The analysis was carried out using images from TCGA. Were these frozen sections or diagnostic samples? If both were used, did authors find differences in spatial organization between them? Authors used a convolutional neural network to generate TIL maps. How was the performance of such a method in the employed dataset?
Response: In this study, we have not generated our own TIL maps, but used the one provided in [Abousamra, et al. "Deep Learning-Based Mapping of Tumor Infiltrating Lymphocytes in Whole Slide Images of 23 Types of Cancer." Frontiers in Oncology 11 (2022)] to test different methods of measuring the spatial organization of TILs. We are aware that this misunderstanding may be related to an imprecise description of the topic in the manuscript, therefore we have changed the following text in the Introduction (last paragraph) and Methods description:
“In this work, we have collected and evaluated 14 different measures of spatial heterogeneity (termed spatial measures), including the one previously applied to TIL maps analysis and also methods developed for the analysis of imaging mass spectrometry or spatial transcriptomics data. We tested these measures on TIL maps from 23 cancer types created elsewhere [12] with the ultimate goal to find the most robust and efficient indices that could be used for cancer prognosis.”
Remark 4: As mentioned before, some sections would benefit from rephrasing. For example, the paragraphs 2 and 3 of Section 2.2 are not completely clear. For example, the paper says “Binarizing the TIL map into regions with and without TILs allowed to separate them into regions-of-interest”; this sounds confusing, does it mean that there are regions of interest for TILs and other for non-TILs? Authors are encouraged to include a figure (or adjust Figure 1) that clearly illustrates, step by step, the data processing. Similarly, paragraph 2 of section 3.2 is hard to follow and should provide more details. Also, in that paragraph, please indicate which correlation authors are referring to (Spearman’s or Pearson’s).
Response: Thank you for pointing this out! We extended the description of the methods and added additional supplementary figure 1 to present the data processing step more precisely. Also, we have thoroughly revised paragraphs that are hard to follow in sections 2.2 and 2.3.
Remark 5: The way authors discretized data seems a little unorthodox. Please include some references that back this approach.
Response: We have selected the cutoff value using the minimum p-value approach introduced by Miller and Sigmund for binary outcomes (Miller, Rupert, and David Siegmund. "Maximally Selected Chi-Square Statistics." Biometrics (1982): 1011-16.) but also applied in survival data analysis with some modifications (Mazumdar, Madhu, and Jill R Glassman. "Categorizing a Prognostic Variable: Review of Methods, Code for Easy Implementation and Applications to Decision‐Making About Cancer Treatments." Statistics in medicine 19, no. 1 (2000): 113-32.). We added the following sentences to the manuscript to extend the description (section 2.4):
“To dichotomize continuous predictors in survival analysis, a minimum p-value approach was used [18]. The method was developed to select a cut-point with a maximum χ2 statistic when the outcomes are binary but was also extended to survival outcomes [19]”.
Remark 6: Statistical significance level was set to 0.05. Were p-values two-sided?
Response: Yes, in all statistical tests we used two-sided tests. We added the following sentence to the manuscript (section 2.5):
“Two-sided tests were performed in all cases.”
Remark 7: According to Figure 1, some images contained consecutive cuts while others had different tissue fragments, and the experiment described in section 3.3 studied the reproducibility of spatial measures across ROIs. It sounds expected that there is correlation between such reproducibility and the type of ROI. A higher reproducibility should be found when consecutive cuts are analyzed as opposed to different fragments. Authors need to further discuss this point.
Response: That’s a good point worth analyzing. Unfortunately, we do not have the information if multiple ROIs for the same patient are consecutive cuts or different tissue fragments since both conditions could happen on a single slide. The manual investigation of each of the over 10,000 ROIs could be imprecise and biased, therefore, we have only added the following text in the discussion:
“Lastly, in some WSIs consecutive cuts of the same tissue were present while in the others different tissue fragments were stored. We expect that a higher reproducibility should be found when consecutive cuts are analyzed as opposed to different fragments, but due to lack of information on which ROIs are consecutive cuts, we could not check it.”
Remark 8: The paper says that “three measures with the highest cancer prognosis capability were GLCM M1, Ripley’s L, and spatial autocorrelation, having similar position in ranking for different endpoints and Cox model types. Across cancers, the highest correlation of spatial measures with survival was observed for hnsc and skcm”. How did authors measure the prognostic capabilities of the models? If that is based in p-value, this may not be correct (See previous comment). Please clarify.
Response: Thank you for the comment! As you can find in the response to remark 1, we calculated Harrell’s concordance index and use it to compare the results between spatial measures and cancer types. We introduced the description of new findings in section 3.2 (as shown in response to remark 1).
Remark 9: Although the volcano plots look nice, they do not provide much information since the cancer type cannot be determined. The plots presented in Supplementary Figure 6 look more interesting in this sense, so they should be included in the main manuscript.
Response: We are sorry that this figure is not informative enough for the reader. As you suggested, we replaced the Volcano plot with some plots for supplementary figure 6 and also added Harrell’s concordance index across cancers.
Remark 10: Discussion should include some recent works in spatial analysis of TILs in different cancer types such as oropharyngeal and ovarian (See for example 10.1093/jnci/djab215 and 10.1136/jitc-2021-003833). Also, authors need to comment on the differences between their approach and the others existent in the literature.
Response: Thank you for the comment. We added these two papers into the bibliography and the following sentences into the Discussion:
“Another measure was developed to characterize the spatial architecture patterns of TILs together with surrounding cells for HPV-associated oropharyngeal squamous cell carcinoma and was associated with DFS in low-risk patients [23]. ArcTIL was created for quantitative characterization of the architecture of TILs and their interplay with cancer cells in three different gynecological cancers [24]. Most of these measures were developed exclusively for a given cancer; here we reviewed measures in terms of universal prediction models for multiple cancers. Also, the methods described above are much more complicated and might include additional covariates or clinical features to be able to calculate, while we assumed that TIL maps should be enough.”
Remark 10: Finally, Discussion must include the limitations of the work.
Response: The paragraph describing the limitations of the study was added to the manuscript:
“The study has some limitations. First, in the analysis we used TIL maps that were provided by others [12]. In the original study, they showed high concordance of TIL maps with expert annotations, however, they might not be perfect in some cases. The spatial measures calculated on different TIL maps might bring slightly different survival results, however, we assume that the average overall ranking should be similar. Another limitation is that for some measures it is possible to get better results by tuning their parameters; here we used default values. The next limitation is that survival results depend on the method used for dichotomization of each measure into low/high groups and the minimum p-value algorithm used here might not be optimal in all cases. Lastly, in some WSIs consecutive cuts of the same tissue were present while in the others different tissue fragments were stored. We expect that a higher reproducibility should be found when consecutive cuts are analyzed as opposed to different fragments, but due to lack of information on which ROIs are consecutive cuts, we could not check it.”
Reviewer 2 Report
Major
-
Why the probability value from 0.5 to 1 instead of 0 to 1? Any reference for the decision?
-
How did we decide the penalized parameter to be 0.1 in line 159? Is it based on cross validation?
-
When we average the spatial measurement by ROI of the same patient, did we weighted by the ROI size (like total number of pixels)?
-
It’s natural that when dichotomizing the value to high/low, we are reducing the variants in individual groups and increasing the separation. I am a little concerned about how the low/high groups are decided and whether the cutoff is robust. Can the authors provide a bit deeper analysis on this?
-
Is it possible for the authors to draw some more concrete conclusions or suggestions?
Minor
-
Weird sentence line 60-63
-
Please include all the analysis code in a public repository (like github) or in the supplement for replication purposes.
Author Response
Remark 1: Why the probability value from 0.5 to 1 instead of 0 to 1? Any reference for the decision?
Response: That’s a good point to clarify. The TIL maps that were used in this study, were generated by the Authors of [Abousamra S, et al. Deep Learning-Based Mapping of Tumor Infiltrating Lymphocytes in Whole Slide Images of 23 Types of Cancer. Frontiers in Oncology (2022) 11] and the files provided by them had only values from 0.5 to 1. As they state in their manuscript, they set the threshold to 0.5 and if the probability value was less than 0.5, the patch is not representing TIL and is treated as tissue or background. We added the following sentence to manuscript:
“In the original paper [12], the information about TIL was stored using two TIL map scales: (i) binary (if there are TILs on the patch or not); (ii) probability (values from 0.5 to 1; closer to 1 means a higher chance of TILs; values smaller than 0.5 means that there are no TILs on the patch)”
Remark 2: How did we decide the penalized parameter to be 0.1 in line 159? Is it based on cross validation?
Response: Thank you for this comment. Since there were 16 different scenarios of survival analysis for 14 spatial measures in 23 cancer types, we were not able to find the one, optimal value of the penalized parameter. We selected the value 0.1 as a good compromise for all cases.
Remark 3: When we average the spatial measurement by ROI of the same patient, did we weighted by the ROI size (like total number of pixels)?
Response: That’s a good point. In this analysis, we have not weight the measures by the ROI size during averaging, since we assume that they should be independent of the ROI size. If the spatial co-localization is the same on the bigger and smaller fragment of the tissue, a good spatial measure should give the same value.
Remark 4: It’s natural that when dichotomizing the value to high/low, we are reducing the variants in individual groups and increasing the separation. I am a little concerned about how the low/high groups are decided and whether the cutoff is robust. Can the authors provide a bit deeper analysis on this?
Response: We have selected the cutoff value using the minimum p-value approach introduced by Miller and Sigmund for binary outcomes (Miller, Rupert, and David Siegmund. "Maximally Selected Chi-Square Statistics." Biometrics (1982): 1011-16.) but also applied in survival data analysis with some modifications (Mazumdar, Madhu, and Jill R Glassman. "Categorizing a Prognostic Variable: Review of Methods, Code for Easy Implementation and Applications to Decision‐Making About Cancer Treatments." Statistics in medicine 19, no. 1 (2000): 113-32.). We are aware, that a better method for finding the cutoff exists, however, we wanted to choose a simple method that could be calculated for any spatial measure. We added the following sentences to the manuscript:
“To dichotomize continuous predictors in survival analysis, a minimum p-value approach was used [18]. The method was developed to select a cut-point with a maximum χ2 statistic when the outcomes are binary but was also extended to survival outcomes [19]”.
Remark 5: Is it possible for the authors to draw some more concrete conclusions or suggestions?
Response: We thoroughly extended the Discussion and Conclusions part in the revised version of the manuscript.
Remark 6: Weird sentence line 60-63
Response: The sentence was modified. Hopefully, it is now clear for the reader.
“In this work, we have collected and evaluated 14 different measures of spatial heterogeneity (termed spatial measures), including the one previously applied to TIL maps analysis and methods developed for the analysis of imaging mass spectrometry or spatial transcriptomics data. We tested these measures on TIL maps from 23 cancer types with the ultimate goal to find the most robust and efficient indices that could be used for cancer prognosis.”
Remark 7: Please include all the analysis code in a public repository (like github) or in the supplement for replication purposes.
Response: As you pointed out, to allow reproducibility of the study we added the code for data processing, calculating spatial measures, and survival analysis on our Github webpage. Also, we included this information in the manuscript:
“Computational pipeline developed for TIL maps processing, calculation of spatial measures and survival analysis is freely available for download from https://github.com/Aleksandra795/TIL_maps.”
Round 2
Reviewer 2 Report
I appreciate the authors' effort to address my comments and concerns. One minor follow up, can the authors incorporate remark 2 and remark 3 to the manuscript in case other readers have similar question?
Author Response
General comments: I appreciate the authors' effort to address my comments and concerns. One minor follow-up, can the authors incorporate remark 2 and remark 3 to the manuscript in case other readers have similar question?
Response: Thank you for pointing this out. We corrected this mistake by updating the manuscript with the following sentences:
“Since there were 16 different scenarios of survival analysis for 14 spatial measures in 23 cancer types, we were not able to find the one, optimal value of the penalized parameter. The selected value was a good compromise for all cases”.
“We have not weighted the measures by the ROI size during averaging, since we assume that they should be independent of the ROI size. If the spatial co-localization is the same on the bigger and smaller tissue fragment, a good spatial measure should give the same value.”